# A Broad Characterization of Glycogen Storage Disease IV Patients: A Clinical, Genetic, and Histopathological Study

**DOI:** 10.3390/biomedicines11020363

**Published:** 2023-01-26

**Authors:** Matheus Vernet Machado Bressan Wilke, Bibiana Mello de Oliveira, Rodrigo Tzovenos Starosta, Marwan Shinawi, Liang Lu, Mai He, Yamin Ma, Janis Stoll, Carolina Fischinger Moura de Souza, Ana Cecilia Menezes de Siqueira, Sandra Maria Gonçalves Vieira, Carlos Thadeu Cerski, Lilia Farret Refosco, Ida Vanessa Doederlein Schwartz

**Affiliations:** 1Medical Genetics Service, Hospital de Clínicas de Porto Alegre, Ramiro Barcelos St., 2350, 3rd Floor, Porto Alegre 90035-903, Brazil; 2Post Graduation Program Ciências da Saúde, Universidade Federal do Rio Grande do Sul, Porto Alegre 90035-903, Brazil; 3Post Graduation Program in Genetics and Molecular Biology, Universidade Federal do Rio Grande do Sul, Porto Alegre 90035-903, Brazil; 4Division of Medical Genetics and Genomics, Washington University in Saint Louis, Saint Louis, MO 63130, USA; 5Department of Pathology and Immunology, Washington University in Saint Louis, Saint Louis, MO 63130, USA; 6Division of Pediatric Gastroenterology, Hepatology, and Nutrition, Washington University in Saint Louis, Saint Louis, MO 63130, USA; 7Postgraduate Program in Child and Adolescent Health, Universidade Federal do Rio Grande do Sul, Porto Alegre 90035-903, Brazil; 8Treatment Center of Inborn Errors of Metabolism, Instituto de Medicina Integral Professor Fernando Figueira, Recife 50070-902, Brazil; 9Pediatrics Gastroenterology Service, Hospital de Clínicas de Porto Alegre, Porto Alegre 90035-903, Brazil; 10Pathology Service, Hospital de Clínicas de Porto Alegre, Porto Alegre 90035-903, Brazil; 11Department of Genetics, Universidade Federal do Rio Grande do Sul, Porto Alegre 90035-903, Brazil; 12BRAIN Laboratory, Hospital de Clínicas de Porto Alegre, Porto Alegre 90035-903, Brazil

**Keywords:** glycogen storage disease IV, liver transplantation, dietary treatment

## Abstract

Glycogen storage disease type IV (GSD IV) is an ultra-rare autosomal recessive disease caused by variants in the *GBE1* gene, which encodes the glycogen branching enzyme (GBE). GSD IV accounts for approximately 3% of all GSD. The phenotype of GSD IV ranges from neonatal death to mild adult-onset disease with variable hepatic, muscular, neurologic, dermatologic, and cardiac involvement. There is a paucity of literature and clinical and dietary management in GSD IV, and liver transplantation (LT) is described to correct the primary hepatic enzyme defect. **Objectives:** We herein describe five cases of patients with GSD IV with different ages of onset and outcomes as well as a novel *GBE1* variant. **Methods:** This is a descriptive case series of patients receiving care for GSD IV at Reference Centers for Rare Diseases in Brazil and in the United States of America. Patients were selected based on confirmatory *GBE1* genotypes performed after strong clinical suspicion. **Results:** Pt #1 is a Latin male with the chief complaints of hepatosplenomegaly, failure to thrive, and elevated liver enzymes starting at the age of 5 months. Before LT at the age of two, empirical treatment with corn starch (CS) and high protein therapy was performed with subjective improvement in his overall disposition and liver size. Pt #2 is a 30-month-old Afro-American descent patient with the chief complaints of failure to gain adequate weight, hypotonia, and hepatosplenomegaly at the age of 15 months. Treatment with CS was initiated without overall improvement of the symptoms. Pt #3.1 is a female Latin patient, sister to pt #3.2, with onset of symptoms at the age of 3 months with bloody diarrhea, abdominal distention, and splenomegaly. There was no attempt of treatment with CS. Pt #4 is an 8-year-old male patient of European descent who had his initial evaluation at 12 months, which was remarkable for hepatosplenomegaly, elevated ALT and AST levels, and a moderate dilatation of the left ventricle with normal systolic function that improved after LT. Pt #1, #3.2 and #4 presented with high levels of chitotriosidase. Pt #2 was found to have the novel variant c.826G > C p.(Ala276Pro). **Conclusions:** GSD IV is a rare disease with different ages of presentation and different cardiac phenotypes, which is associated with high levels of chitotriosidase. Attempts of dietary intervention with CS did not show a clear improvement in our case series.

## 1. Introduction

Glycogen storage disease type IV (GSD IV—OMIM #232500) is a hepatomuscular inborn error of the metabolism caused by a deficiency of glycogen branching enzyme (E.C. 2.4.1.18), encoded by *GBE1*. This deficiency leads to the accumulation of structurally abnormal glycogen, which is similar to amylopectin (polyglucosan) in hepatocytes, myocytes, and neurons. Incidence is estimated at 1:600,000 to 1:800,000 individuals worldwide [1]. GSD IV can be classified according to age of onset and phenotype as a classic hepatic form, a non-progressive hepatic form, and a neuromuscular form [2,3]. Biallelic pathogenic variants in *GBE1* are also associated with adult polyglucosan body disease (APBD), which is characterized by the accumulation of spherical intra-axonal inclusion bodies in the central and peripheral nervous systems, leading to mixed central and peripheral neurological involvement without liver damage. This corroborates GSD IV being a pleiotropic disorder with a broad phenotypic continuum [3,4].

As there is no specific treatment available for GSD IV, medical care is primarily focused on managing symptoms and promoting quality of life. Individuals with progressive liver disease benefit from liver transplantation (LT) when there is acute liver failure or decompensated liver cirrhosis. Long-term post-transplantation care goals include surveillance for involvement of the heart, an important extra-hepatic site of amylopectin accumulation [1]. A recent study suggested that dietary management may have the potential to improve clinical and laboratory outcomes and delay or prevent liver transplantation [3]. Newer treatment modalities, including gene therapy, are currently being investigated with promising results in animal models [5].

In this article, we describe the clinical manifestations of five individuals diagnosed with GSD IV and analyze aspects of natural history, dietary management, cardiac phenotype, and outcomes of liver transplantation.

## 2. Methods

This is a descriptive case series of patients receiving care for GSD IV at Reference Centers for Rare Diseases in Brazil and in the United States of America. Patients were selected based on confirmatory *GBE1* genotypes performed after strong clinical suspicion. Data on sociodemographic characteristics, clinical history, histopathological findings, and genetic analysis information were retrieved from medical records.

## 3. Results

### 3.1. Case 1

A two-year-old Latin male child, from the southeast region of Brazil, presented with severe hepatosplenomegaly, failure to thrive, elevated liver enzymes, and synthetic liver dysfunction. He is the third child of non-consanguineous parents. He has two healthy siblings, and the family history was noncontributory. From the neonatal period to five months of age, his developmental and nutritional status was within normal range. At five months of age, the patient was admitted with skin lesions suggestive of hand–foot–mouth disease. His disease course was complicated by pleural and pericardial effusions, requiring admission to the Intensive Care Unit. After discharge, he developed jaundice, ascites, persistently high liver transaminases (ALT, AST), and γ-glutamyl transpeptidase (GGT), evolving to progressive chronic liver dysfunction around the age of 9 months. The patient was transferred to our institution at one year and eleven months of age for further evaluation and consideration of liver transplantation. At admission, at the age of 23 months, the patient was underweight (weight 9.7 kg, −2.42 SD; length 82 cm, −1.28 SD, DC Growth Charts). Physical examination revealed firm hepatomegaly (liver palpable at 5 cm below right costal margin), splenomegaly (spleen palpable at 10 cm below left costal margin), and hypotonia. The results of biochemical and metabolic tests are shown in Table 1. An abdominal Doppler ultrasound showed an enlarged liver with heterogeneous echogenicity, splenomegaly, and grade 1 ascites. An echocardiography showed a left ventricle diameter in the upper limits of the reference range for age. A bone marrow biopsy with immunophenotyping was normal. Metabolic investigation included increased chitotriosidase and ferritin levels: 860 nmol/h/mL (NRV = 8.8–132) and 375.6 ng/mL (NRV = 21–274), respectively. Glucocerebrosidase and acid sphingomyelinase activities in leukocytes were 6.2 nmol/h/mg prot (NRV = 10–45) and 0.3 nmol/h/mg (NRV = 0.7–4.9), respectively; however, poor sample integrity was noted, and those results were considered false-positive due to pre-analytical factors. Lyso-sphingomyelin-509 (Lyso-509) levels and targeted analysis of *NPC1* and *NPC2* were normal. A liver biopsy showed bridging fibrosis with thick septa (Laennec grade 4C) with regenerative nodule organization; hepatocytes had intracellular hydropic degeneration and cytoplasmic inclusions, which were positive for periodic acid-Schiff (PAS) and resistant to diastase treatment, compatible with glycogen storage disease (GSD) (Figure 1B,C). A commercial next generation sequencing (NGS) panel (Panel of Treatable Diseases—Mendelics Genomic Analysis Laboratory, including 340 genes) identified two variants in *GBE1:* c.1544G > A (p.Arg515His), classified as pathogenic; and c.1064G > A (p.Arg355His, NM_000158), classified as likely pathogenic and confirmed to be *in trans* by segregation analysis. The diagnosis of GSD type IV was established. Due to cirrhosis, the patient was listed for liver transplantation with a Pediatric End-Stage Liver Disease (PELD) score of 18 and CHILD Pugh C classification.

An attempt to improve the patient’s nutritional status using cornstarch (CS) at 0.8 g/kg every 6 h was implemented after the diagnosis was established. CS was increased as tolerated to 1.17 g/kg every 6 h. The patient was also started on a high protein diet (3 g/kg/day). After the start of the therapy, there was subjective improvement in his overall disposition, endurance, and liver size noticed by his relatives, although no objective parameter of improvement could be measured. Despite treatment, two months after hospital discharge there was worsening of ascites and increases of ALT, AST, and GGT. The patient received a related living-donor liver transplant from his maternal aunt at 27 months old. Images of the explanted liver are shown in Figure 1A. The post-surgical course was complicated by biliary stenosis requiring a new anastomosis. The patient’s siblings were heterozygous for the *GBE1* variants on targeted analysis.

### 3.2. Case 2

A 30-month-old female patient of Afro-American descent was referred for genetic evaluation at the age of 15 months with the chief complaints of failure to gain adequate weight, hypotonia, and hepatosplenomegaly. The patient was born to non-consanguineous parents. Pregnancy was complicated by oligohydramnios, which led to a C-section delivery at 36 weeks. The mother is a carrier of alpha thalassemia, but the family history was otherwise noncontributory. To investigate the hepatosplenomegaly, the abdominal MRI was performed showing hepatomegaly with a lobular surface suggestive of cirrhosis, ascites, and numerous lesions consistent with dysplastic nodules. A liver biopsy at 16 months old showed features of liver storage disease and portal and periportal bridging fibrosis (METAVIR stage F3–4) (Figure 1D,E). An echocardiogram at 25 months old was normal. Laboratory testing results are summarized in Table 1. There was no report of hypoglycemia. Genetic testing using a commercial NGS panel (Invitae Comprehensive Glycogen Storage Disease Panel) identified two variants in *GBE1*: c.1544G > A; p.(Arg515His) classified as pathogenic, and c.826G > C; p.(Ala276Pro) classified as a VUS. Segregation was not performed in this case. Treatment with 1 g/kg CS nightly was initiated without an overall improvement of the symptoms. The patient underwent liver transplantation from a deceased donor at the age of 30 months. The post-operative course was uncomplicated; she is currently 34 months old and is catching up on growth and muscle tone.

### 3.3. Cases 3.1 and 3.2

Cases 3.1 and 3.2 are siblings from the northeast region of Brazil. Pt #3.1 is a Latin female patient who presented at 3 months old with bloody diarrhea, abdominal distention, and hepatosplenomegaly. The patient was born to non-consanguineous parents and had an uncomplicated pregnancy. An abdominal ultrasound at presentation showed an enlargement of spleen and liver, with a homogeneous echotexture. There was no hypoglycemia, anemia, or thrombocytopenia. A commercial NGS gene panel (Panel of Treatable Diseases—Mendelics Genomic Analysis Laboratory) at 6 months old showed two pathogenic variants in the *GBE1* gene: c.1544G > A; p.(Arg515His) and c.1803 + 2T > C; p.?. Segregation of these variants was not performed. CS therapy was not attempted. The patient is currently listed for LT at 6 months old.

Patient #3.2, patient #3.1′s older brother, died at eighteen months of age with the diagnosis of hepatosplenomegaly and histiocytosis. He underwent a metabolic investigation that included normal beta-glucosidase, beta-galactosidase, and lysosomal acid lipase activity despite an elevated chitotriosidase activity at 973 nmol/h/mL (NRV = 8.8–132). Bone marrow biopsy showed findings characteristic of histiocytosis. The liver biopsy was retrieved due to patient 3’s diagnosis with suggestive GSD IV findings, including hepatic fibrosis and the presence of hepatic nodules. No information about PAS was available, and no molecular testing was pursued before death.

### 3.4. Case 4

An 8-year-old male of Northern European descent was born to non-consanguineous parents and was evaluated at 12 months of age at an outside hospital due to underweight despite adequate caloric intake. He was born at 37 weeks of gestation after an uncomplicated pregnancy and delivery and had transient hypoglycemia in the first day of life, which was responsive to nutritional support. His initial physical examination was remarkable for hepatosplenomegaly, and laboratory studies showed elevated ALT and AST levels as well as TSH and T4 levels suggestive of hypothyroidism. Levothyroxine was prescribed without improvement of growth parameters. At 19 months old, the patient presented with severe anemia (Hb 3.5 g/dL, NRV = 11.0–13.5 g/dL) and was referred to our institution for further evaluation and management. At this time, the patient was noticed to have gross motor delays with adequate fine motor and communication skills. A chromosomal microarray and a metabolic investigation including serum amino acid profile, urine organic acid profile, acylcarnitine profile, lysosomal enzyme panel, and carbohydrate-deficient transferrin were unremarkable. Abdominal ultrasound showed a mildly enlarged liver (10.7 cm in length, normal <10.0) and spleen (8.6 cm in length, normal <8.0) with normal echotextures. An upper endoscopy was normal, and a colonoscopy was significant for several colonic friable vascular lesions, which were not biopsied due to coagulopathy and low hemoglobin levels. An echocardiogram showed moderate dilatation of the left ventricle (Z score = +5.7) with normal systolic function. A liver biopsy showed marked alteration of the lobular architecture with portal to portal and intralobular sinusoidal fibrosis, with borderline cirrhosis criteria (Laennec 4A); the hepatocytes contained hyaline inclusions, which were positive for PAS, PAS-D, and colloidal iron staining. Branching enzyme activity in the liver biopsy sample was undetectable (NRV > 5.4 μmol/min/gram tissue), and glycogen analysis studies showed a low glycogen content at 1.3 μmol of glucose/mg tissue, with a normal glucose-1-phosphate-to-glucose at 21.94% (NRV > 5%). Commercial targeted *GBE1* sequencing by NGS (Invitae Comprehensive Glycogen Storage Disease Panel) at 21 months old showed compound heterozygous c.1570C > T, p.(Arg524*)/c.760A > G, p.(Thr254Ala) classified as pathogenic and likely pathogenic, respectively. The clinical course as well as molecular and biochemical data were consistent with GSD IV, and the patient was listed for liver transplantation. At 24 months old, the patient received a deceased-donor liver transplantation. The patient is currently 8 years old and has no signs of portal hypertension. He continues to be at low percentiles for weight (last measurement: CDC percentile <1%) and height (last measurement: CDC percentile 4.9%). Currently, he eats a normal diet, his hypoglycemia and left ventricular dilatation are resolved, and he is caught up on his motor development.

## 4. Discussion

GSD IV is a rare metabolic disease that may lead to the accumulation of dysfunctional glycogen in hepatocytes, skeletal and cardiac myocytes, and neural cells. In this article, we presented a retrospective case series with four previously unpublished cases of GSD IV—three of which underwent LT—with molecular confirmation and a “classic”, progressive hepatic phenotype, in addition to a patient presumed to have GSD IV but who did not have molecular testing.

Liver transplantation is indicated for GSD IV patients with the progressive hepatic phenotype when they exhibit end-stage liver disease with synthetic failure and portal hypertension. Given the rarity of this disorder, outcomes of LT are mainly reported non-systematically as single case reports. A recent review of 24 patients with GSD IV who underwent LT showed an overall mortality of 37.5% after LT compared to virtually 100% in untreated end-stage liver disease patients with GSD IV [6]. Cardiomyopathy one-month post-liver transplantation and sepsis ranging from 7 days to 0.3 years post-transplantation were appointed as the most common causes of mortality, after LT, and there was an overall improvement of growth parameters and of cardiac disease [1]. Although there is great variability in the reported cardiac responses to LT in the literature, there have been comparisons of pre- and post-LT myocardial biopsies showing decreases in cardiac glycogen storage after LT [6,7]. It has been demonstrated that the migration of cells from the allograft (microchimerism) could explain the improved metabolism of enzyme-deficient tissues in the recipient as it was demonstrated by polymerase-chain-reaction analysis in the heart of patients with GSD [8]. The progression of neurological disease after LT has only been documented in one patient [9].

Dilated cardiomyopathy is the most common manifestation of cardiac disease in GSD IV. Functional assays using animal models have shown that *GBE* ablation led to fetal hydrops and embryonic lethality through ventricular non-compaction [10]. Although there are currently no official guidelines for the follow-up of individuals with GSD IV, periodical echocardiograms are generally recommended throughout life [2]. In our case series, only one out of five patients had signs of cardiomyopathy, which fully reversed after LT; however, all patients in our cohort are still young and will continue to be monitored on a consistent basis for development of heart disease.

A recent observational study by Derks et al. (2021) evaluated 15 individuals with GSD IV and suggested that dietary management, as performed in many other GSDs, may have the potential to delay or prevent liver transplantation, improve growth, and normalize serum aminotransferases in these patients [3]. In the present series, empirical dietary intervention was performed in two cases, without clinical success.

The GBE structure contains a conserved amylase core that harbors the active site for the branching reaction and a catalytic domain encompassing amino acid residues 184 to 600, where all variants from the cases would be located [11,12].

Three out of four patients with the genotype available carry the variant c.1544G > A in the compound heterozygous state. This variant is found in 10 alleles of 275,166 (frequency: 0.0036%) in gnomAD from different populations. This variant was previously reported in association with both APBD and GSD IV. In one 46-year-old female patient with gait disturbance and urinary urge incontinence, the variant was found in a compound heterozygous state with p. (Arg524Gln). Postmortem assessment of this case showed numerous periodic acid Schiff–positive polyglucosan bodies (PB) found throughout the brain and spinal cord, myocardium, and peripheral nerve but not in skeletal muscles or liver [13]. The same variant was found in a homozygous state in a female patient presenting with hepatomegaly and elevated transaminase levels at the age of 2 [14]. Her liver biopsy demonstrated increased glycogen content. Her liver disease resolved spontaneously, and she started demonstrating additional symptoms of APBD at the age of 44 years, including difficulty in writing and walking due to weakness and stiffness. Sural nerve biopsy revealed typical intracytoplasmic PB inclusions in neuronal axons. Another patient compound heterozygous with c.1961_1962delC presented at the age of 37 years with episodes of leg dragging and slurred speech, and his residual glycogen branching enzyme activity was 30% [14].

The c.1544 G > A variant was already associated with GSD IV in a patient who was compound heterozygote with a c.1612T > G and who presented with recurrent fever and jaundice since 9 months of age [15]. Patient #2 had this variant in a heterozygous state (*in trans* not confirmed) with the novel c.826G > C; p.(Ala276Pro) variant. This variant is in silico deleterious with a Revel score of 0.93, being absent in gnomAD. We believe this variant could be further elevated to likely pathogenic after phasing both variants due to the high specificity of the phenotype.

Hemophagocytic lymphohistiocytosis (HLH) was identified in patient#3.2. Inborn errors of metabolism can be associated with secondary HLH in children being probably related to the accumulation of nondegraded substrates in macrophages that may lead to uncontrolled macrophage activation and the subsequent development of HLH [16]. Thus, the finding of HLH in these patients may reflect a complication of GSD IV [16]. On the other hand, several infectious agents such as Epstein–Barr virus and cytomegalovirus may trigger a hemophagocytic response. To assign this finding as a GSD IV complication or even a primary HLH primary or virus associated is not possible.

Missense variants in *GBE1* are described to be associated with milder forms such as non-progressive GSD and APBD, making a genotype–phenotype relationship less clear for these variants [12]. The variant p. Arg355His, present in patient #1, was also found in a compound heterozygous state with c.1604A > G in three Italian siblings presenting in their fifties with a combination of pyramidal and ataxic signs and mild demyelinating neuropathy [17]. Both muscle and nerve biopsies showed polyglucosan bodies in the siblings with polyneuropathy. The genotype of patient #1 was reported in one patient with GSD IV in an exome sequencing cohort [18]. Long-term surveillance for neurological symptoms in patients with GSD IV will be important to better understand the clinical spectrum of this disorder.

Patient #3.1 had a variant affecting splicing (c.1803 + 2T), and she appeared to have a more severe phenotype, including a younger age of onset than the rest of the patients in our series. The c.1803 + 2T > C variant was recently reported as being causative for GSD IV. This is a null variant located between exons 13 and 14 in a conserved nucleotide. This variant is predicted to undergo nonsense mediated decay in a gene where LOF is a known disease mechanism. This variant is present in gnomAD in 1 allele of 224,308 and in 0 homozygotes (frequency: 0.0004%). According to the literature, null variants, with the exception of those present in exons 15 and 16, have resulted in more severe forms, including classic hepatic and congenital neuromuscular forms, in the homozygous state [12]. The variants in patient #4 were already described in the literature as being associated with classic and juvenile GSD IV [12].

Interestingly, three of the patients (patients #1,#3.2 and #4) in this cohort had elevated blood chitotriosidase levels. Chitotriosidase is a marker of macrophagic activation used in diagnosis and follow-up of patients with some lysosomal storage disorders, especially the sphingolipidoses [19]. This finding is in line with the previous reports of elevated chitotriosidase levels in patients with a severe hepatic phenotype of GSD IV and highlights the need for including GSD IV in the differential diagnosis of hepatosplenomegaly with elevated chitotriosidase levels. Larger studies are needed to investigate a possible role of chitotriosidase as a biomarker for GSD IV [16,20].

In conclusion, we present a case series of patients with GSD IV with different ages of presentation, cardiac phenotypes, and attempts of dietary intervention to further illustrate the phenotypic spectrum and therapeutic approaches of this rare disease. Three patients presented with high levels of chitotriosidase, and only one patient had a clear cardiac involvement by the time of LT. To the best of our knowledge, this is the first time that the variant c.826G > C; p.(Ala276Pro) has been described in association with the GSD IV phenotype. Follow-up of these cases will help to better improve treatment recommendations and guidelines for managing and diagnosing patients with GSD IV.

## Figures and Tables

**Figure 1 biomedicines-11-00363-f001:**
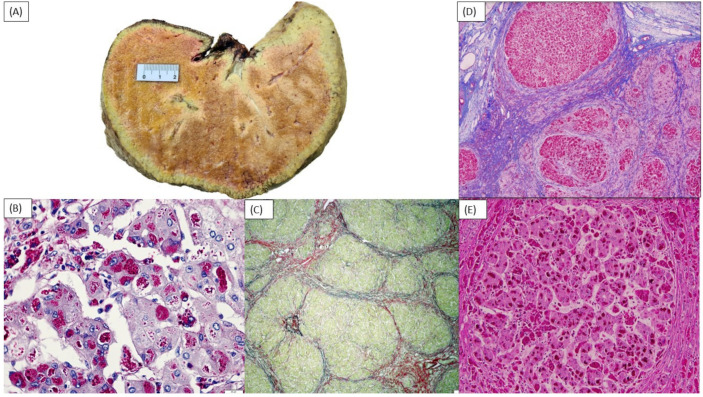
Macro and micro aspects of the explanted liver of Case 1 (**A**–**C**) and Case 2 (**D**,**E**). (**A**) Patient’s liver showing accumulation of amylopectin. Ruler shows the measurement in cm. (**B**) Hematoxylin and eosin staining showing intracellular hydropic degeneration and cytoplasmic inclusions, and (**C**) periodic acid-Schiff-diastase (PAS-D) staining. Hepatocytes were stained with PAS and were mostly digested by PAS-D. (**D**) 20 × magnification: Trichrome stain shows features of liver storage disease and portal and periportal bridging fibrosis. (**E**) 100 × magnification: Hematoxylin and eosin staining showing intracellular hydropic degeneration and cytoplasmic inclusions.

**Table 1 biomedicines-11-00363-t001:** Patients’ demographic and clinical findings.

	Pt# 1	Pt #2	Pt #3.1	Pt #3.2	Pt #4
**Ethnic Origin**	Latino	African American	Latino	Latino	Northern European descent
**Sex**	M	F	F	M	M
**Parental consanguinity**	No	No	No	No	No
**Age of onset (months)**	9	15	3	Unknown	12
**Variants (GBE1 gene, NM_000158.4)**	c.1544G > A; p.(Arg515His)/ c.1064G > A/p.(Arg355His)	c.1544G > A; p.(Arg515His)/c.826G > C; p.(Ala276Pro)	c.1544G > A; p.(Arg515His)/c.1803 + 2T > C; p.?	NA	c.1570C > T p.(Arg524*)/ c.760 A > G p.(Thr254Ala)
**Method of diagnosis**	NGS, gene panel	Gene panel	NGS, gene panel	Retrospective (brother of patient 3.1)	NGS, single gene
**Prenatal/neonatal findings**	Normal	Oligohydramnios	Normal	NA	Normal
**Developmental delay**	Yes	Yes	NA	NA	Yes
**Failure to thrive**	Yes	Yes	NA	NA	Yes
**Hypotonia**	Yes	Yes	NA	NA	Yes
**Length in cm (Z score)**	82 (−1.2 SD) at 23 months	NA	65 (−2.18 SD) at 10 months	73 (−2.34 SD) at 16 months	
**Weight in kg (Z score)**	9.7 (−2.4 SD) at 23 months	NA	6.6 (2.69 SD) at 10 months	11.2 (−0.09 SD) at 16 months	
**First symptoms**	Failure to thrive, hepatosplenomegaly, and hypotonia.	Progressive chronic liver dysfunction.	Blood stool/abdominal distension	Hepatosplenomegaly	Hepatosplenomegaly
**Hepatomegaly**	Yes	Yes	Yes	No	Yes
**Splenomegaly**	Yes	Yes	Yes	Yes	Yes
**Serum laboratory data at hospital admission**					
**Chitotriosidase (nmol/h/mL)** **NRV = 8.8–132**	860	N/A	NA	973.2	229
**Hemoglobin (g/dL)** **NRV = 13.2–16.6**	7.1	9.4	12	10.7	7.3
**Platelet count (×10^9^/L)** **NRV = 135–317**	8.9	71	504	NA	264
**Alanine aminotransferase (U/L)** **(NRV = 5–50)**	85	187	999	123	138
**Aspartate aminotransferase (U/L)** **(NRV = 10–60)**	400	271	610	711	405
**Gamma-glutamyl transferase (GGT) (NRV = 7–32)**	152	74	599	380	72
**Lactic acid (NRV < 2.0)**	1.8	5.3	NA	NA	2.4
**US findings**	Hepatomegaly heterogeneous echogenicity.	Hepatomegaly with lobular surface suggesting cirrhosis. Dysplastic nodules; ascites.	Homogeneous hepatosplenomegaly.	NA	Enlarged liver (10.7 cm in length) and spleen (8.6 cm in length) with normal echotextures.
**Echocardiogram findings**	Left ventricle diameter in the upper limits of the normal reference range for age.	Normal	NA	NA	Moderate dilatation of the left ventricle (Z score = +5.7) with normal systolic function. Resolved after LT.
**Liver biopsy findings**	Bridging fibrosis with thick septa (Laennec grade 4C) with regenerative nodule organization; features of storage disease.	Portal and periportal bridging fibrosis (stage 3/4). Histology shows features of storage disease.	NA	Hepatic fibrosis. Lobular Langerhans cell infiltration was not observed.	Findings are bordering on cirrhosis. Inclusions positive for PAS and PAS-D. The inclusions strongly positive for colloidal iron.

NA—not available.

## Data Availability

If necessary, further information is available from the corresponding author upon reasonable request. Further data sharing is not applicable to this article, as no datasets were generated or analyzed during the current study.

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
