# Peer review of "A Broad Characterization of Glycogen Storage Disease IV Patients: A Clinical, Genetic, and Histopathological Study"

_biomedicines, 2023, doi:10.3390/biomedicines11020363_

Round 1

Reviewer 1 Report

I read with great interest your manuscript on a descriptive case series of patients with GSD IV.  Especially the finding of the elevated chitotriosidasis is important to be published and give rise to further investigations and research.

The clinical description of the cases is comprehensive; in table 1 I suggest to include the age of first symptoms and signs as all patients presented with a classic progressive liver disease; the age of 8 years for the 4th patient can be confounding.    Only in the 4th patient there is a clear, systematic description of all metabolic analyses that were performed; the other diagnoses were made by NGS: In my opinion you should mention the methods of investigation (NGS: metabolic panel? glycogen storage panel? or whole exome sequencing/whole genome sequencing?; the lysosomal gene panel needs also to be described in more detail or a reference should be included).  

In the first patient you performed analysis of glucocerebrosidase enzyme activity and NPC by measuring lyso-sphingomyeline-509: why did you only focus on NPC and not on NP or ASMD type B?

In Table 01 you used several times the abbreviation NA: Non Applicable? or Not Performed?  Please clarify.

The normalisation of motoric dysfunction and cardiopathy after liver transplantation in patients with GD IV has already been described in the literature and the hypothesis is that systemic microchimerism in these transplanted patients can give a positive evolution on the heart and skeletal muscle involvement.  Perhaps you should mention this in the discussion (Moses SW et al (2002) Curr Mol Med 2: 177-188).

On p. 9/11 you should carefully look at the description of the effect of the c.1803+2T variant as this doesn't read well.

Some minor remarks: e.g. 2nd patient of Afro-American descent instead of African American.

Author Response

To

Editors of Biomedicines- Special Issue on "Inherited Metabolic Disorders: From Bench to Bedside”

Dear Sir,

As suggested by the reviewers, we are submitting the changes that have been made to our paper entitled: A broad characterization of Glycogen Storage Disease IV patients: a clinical, genetic and histopathological study.

We hope to have addressed all points raised by the reviewers and look forward to the publication of our manuscript. All changes to the text are highlighted in yellow.

We like to thank the reviewers for their time evaluating our manuscript and providing us with their valuable comments, which definitively improved this manuscript. We would also like to thank you for the attention that has been given to the evaluation of our paper.

Sincerely,

                                                Carolina Fischinger Moura de Souza (corresponding author)

Below are the responses to the Reviewer’s comments and suggestions:

REVIEWER 1

I read with great interest your manuscript on a descriptive case series of patients with GSD IV.  Especially the finding of the elevated chitotriosidasis is important to be published and give rise to further investigations and research.

  • The clinical description of the cases is comprehensive; in table 1 I suggest to include the age of first symptoms and signs as all patients presented with a classic progressive liver disease; the age of 8 years for the 4th patient can be confounding.   

Answer: Thank you for this valuable suggestion. We have modified the `current age` for `age of onset` in table 1. First symptoms are already included in table 1.

  • Only in the 4th patient there is a clear, systematic description of all metabolic analyses that were performed; the other diagnoses were made by NGS: In my opinion you should mention the methods of investigation (NGS: metabolic panel? glycogen storage panel? or whole exome sequencing/whole genome sequencing?; the lysosomal gene panel needs also to be described in more detail or a reference should be included).  

Answer: Thank you for this suggestion. Patients 1 and 3.1 underwent commercial genetic testing via the ` Panel of Treatable Diseases` by the Brazilian laboratory Mendelics (https://testedabochechinha.com.br/painel-de-doencas-trataveis/). We have added this information in the line number 135-137. Patients 2 and 5 underwent commercial testing via the Invitae Comprehensive Glycogen Storage Disease Panel. This information was added to the manuscript in lines 162 and 174.

Regarding the different metabolic analysis performed in each patient, we point out that patients came from different sites with different availabilities of biochemical tests. Patient 1 came from a site which is internationally recognized as a reference center for metabolic diseases.

  • In the first patient you performed analysis of glucocerebrosidase enzyme activity and NPC by measuring lyso-sphingomyeline-509: why did you only focus on NPC and not on NP or ASMD type B?

Answer: Thank you for your question. We also measured the activity of acid sphingomyelinase and this information was included in the text in the lines 130-131.

In Table 01 you used several times the abbreviation NA: Non Applicable? or Not Performed?  Please clarify.

Answer: Thank you for your observation. We have corrected table 1 using for NA as “not available”. Further clarification was included in the table 1’s caption.

  • The normalization of motoric dysfunction and cardiopathy after liver transplantation in patients with GD IV has already been described in the literature and the hypothesis is that systemic microchimerism in these transplanted patients can give a positive evolution on the heart and skeletal muscle involvement.  Perhaps you should mention this in the discussion (Moses SW et al (2002) Curr Mol Med 2: 177-188).

Answer: Thank you for the suggestion. We added this in the discussion in lines. This information was added in lines 294-296.

  • On p. 9/11 you should carefully look at the description of the effect of the c.1803+2T variant as this doesn't read well.

Answer: Thank you for the suggestion. We have improved the description in the lines 288-292.

  • Some minor remarks: e.g., 2nd patient of Afro-American descent instead of African American.

Answer: Thank you for this observation. We have corrected in the lines 49 and 151.

Reviewer 2 Report

An interesting case report with detailed descriptions of four (plus one sibling) patients with GSDIV. 

1. Which in silico tools were used to verify the pathogenicity of the two novel variants c.1803+2T and c.826G>C and how common were they in clinvar? 

2. Did you perform segregation analysis of the pathogenitc variants by analysis of DNA in samples from the parents

3. The Table is very large. Maybe some details already mentioned in the description of the patients can be omitted eg ultrasound findings and echocardiographic findings.

4. Some minor details: What do you mean by "synthetic dysfuncion" of the liver?, At some places the exactness is too exact eg chitotriosidase activity in patient 1. Would be enough with two digits. Case 4: weight CDC percentile 0.03 %? 

5. There are a few mistakes in the text eg repetition of spleen size in the middle of the description of patient 4. 

Author Response

January 5th, 2023

To

Editors of Biomedicines- Special Issue on "Inherited Metabolic Disorders: From Bench to Bedside”

Dear Sir,

As suggested by the reviewers, we are submitting the changes that have been made to our paper entitled: A broad characterization of Glycogen Storage Disease IV patients: a clinical, genetic and histopathological study.

We hope to have addressed all points raised by the reviewers and look forward to the publication of our manuscript. All changes to the text are highlighted in yellow.

We like to thank the reviewers for their time evaluating our manuscript and providing us with their valuable comments, which definitively improved this manuscript. We would also like to thank you for the attention that has been given to the evaluation of our paper.

Sincerely,

                                                Carolina Fischinger Moura de Souza (corresponding author)

Below are the responses to the Reviewer’s comments and suggestions:

REVIEWER 2

  • Which in silicotools were used to verify the pathogenicity of the two novel variants c.1803+2T and c.826G>C and how common were they in clinvar? 

Answer: Thank you for your question.  The c.1803+2T>C variant was not previously reported to the best of our knowledge in the literature. Since the manuscript submission, however, this variant was reported 3 times in ClinVar RCV001825542, RCV000792208, RCV002249502, therefore it is not considered novel anymore. We have corrected it on lines 38 and 305.  This is a Null variant in a gene where loss of function is a known mechanism of disease (PVS1), found in 1 Alleles of 224,308 and 0 homozygote in gnomAD.  The variant c.826 G>C was classified based on Revel score Deleterious (Moderate) (0.93), it is described in one entry in ClinVar as a VUS (2022) by INVITAE. I have added a better description of this variant in lines 271-274.

  1. Did you perform segregation analysis of the pathogenic variants by analysis of DNA in samples from the parents

Answer: Thank you for your question.  Segregation was performed in patient 1 but not performed in patient 3.1 due to financial costs. We have added this information in line number 137 and 173. Patients 2 and 5 underwent testing via the Invitae Comprehensive Glycogen Storage Disease Panel with no segregation. This information was added to the manuscript in lines 161 and 209-210.

  1. The Table is very large. Maybe some details already mentioned in the description of the patients can be omitted eg ultrasound findings and echocardiographic findings.

Answer: Thank you for your suggestion.  We have already corrected some information in the table making it more clear in terms of age of onset and availability of data. We believe the paper would benefit on having a more comprehensive table despite duplicated information from the main text.

  1. Some minor details: What do you mean by "synthetic dysfunction" of the liver?, At some places the exactness is too exact eg chitotriosidase activity in patient 1. Would be enough with two digits. Case 4: weight CDC percentile 0.03 %? 

Answer: Thank you for your observations.  We have used "synthetic dysfunction" to summarize the inability to synthesize proteins of the liver (such as prothrombin time (PT)/international normalized ratio (INR), platelet count and albumin level).

We have corrected the digits and values for the two measures appointed lines 127-128 and 215.

  1. There are a few mistakes in the text eg repetition of spleen size in the middle of the description of patient.

Answer: Thank you for your observations.  This was deleted from line 199.

Reviewer 3 Report

The manuscript by Wilke et al reports four cases of a rare genetic syndrome, glycogen storage disease type IV, which comprises ca 3% of all glycogen storage disease cases. As state in the introduction, such patients exhibit a wide range of pathology from mild symptoms in adulthood to acute severe disease progression in neonatal age leading to patient's death. Sometimes, the patients show signs of liver dysfunction and increase in size. This manuscript described four such patients registered in Brazil and the US centers. A detailed medical history is presented. All four patients indeed carried mutations in the GBE1 gene.

The manuscript is clearly written. Introduction contains all required information. The conclusions are based on the presented information.

Actually, the only drawback of the manuscript is the sole typo in the abstract (monhs).

Author Response

January 5th, 2023

To

Editors of Biomedicines- Special Issue on "Inherited Metabolic Disorders: From Bench to Bedside”

Dear Sir or M’am,

As suggested by the reviewers, we are submitting the changes that have been made to our paper entitled: A broad characterization of Glycogen Storage Disease IV patients: a clinical, genetic and histopathological study.

We hope to have addressed all points raised by the reviewers and look forward to the publication of our manuscript. All changes to the text are highlighted in yellow.

We like to thank the reviewers for their time evaluating our manuscript and providing us with their valuable comments, which definitively improved this manuscript. We would also like to thank you for the attention that has been given to the evaluation of our paper.

Sincerely,

                                                Carolina Fischinger Moura de Souza (corresponding author)

Below are the responses to the Reviewer’s comments and suggestions:

REVIEWER 3

The manuscript by Wilke et al reports four cases of a rare genetic syndrome, glycogen storage disease type IV, which comprises ca 3% of all glycogen storage disease cases. As state in the introduction, such patients exhibit a wide range of pathology from mild symptoms in adulthood to acute severe disease progression in neonatal age leading to patient's death. Sometimes, the patients show signs of liver dysfunction and increase in size. This manuscript described four such patients registered in Brazil and the US centers. A detailed medical history is presented. All four patients indeed carried mutations in the GBE1 gene.

The manuscript is clearly written. Introduction contains all required information. The conclusions are based on the presented information.

Actually, the only drawback of the manuscript is the sole typo in the abstract (monhs).

Answer: Thank you for your observations.  This was corrected in line number 54.

Round 2

Reviewer 1 Report

I read with great interest your revised manuscript and all remarks made before are corrected.

Your revised manuscript "A broad characterization of Glycogen Storage Disease IV patients: a clinical, genetic and histopathological study" is ready for publication.